# DYNAMIC PARTITION MODELS

**Marc Goessling**
Department of Statistics
University of Chicago
Chicago, IL 60637, USA
goessling@galton.uchicago.edu

**Yali Amit**
Departments of Statistics and Computer Science
University of Chicago
Chicago, IL 60637, USA
amit@galton.uchicago.edu

## ABSTRACT

We present a new approach for learning compact and intuitive distributed representations with binary encoding. Rather than summing up expert votes as in products of experts, we employ for each variable the opinion of the most reliable expert. Data points are hence explained through a partitioning of the variables into expert supports. The partitions are dynamically adapted based on which experts are active. During the learning phase we adopt a smoothed version of this model that uses separate mixtures for each data dimension. In our experiments we achieve accurate reconstructions of high-dimensional data points with at most a dozen experts.

## 1 INTRODUCTION

We consider the task of learning a compact binary representation (e.g. Goessling & Amit, 2015). That means we are seeking a parsimonious set of experts, which can explain a given collection of multivariate data points. In contrast to most existing approaches the emphasis here is on finding experts that are individually meaningful and that have disjoint responsibilities. Ideally, each expert explains only one factor of variation in the data and for each factor of variation there is exactly one expert that focuses on it.

Formally, the experts $\mathbb{P}_k$, $k = 1, \ldots, K$, are probability distributions that depend on binary latent variables $\boldsymbol{h}(k)$. The latent state $\boldsymbol{h}$ specifies which experts are active and has to be inferred for each $D$-dimensional data point $\boldsymbol{x}$. The active experts then define a probability distribution $\mathbb{P}$. The goal of representation learning is to train experts such that the conditional likelihood $\mathbb{P}(\boldsymbol{x} \mid \boldsymbol{h})$ of the data given the latent activations is maximized.

We start by describing a simple model family, which forms the basis of our work. A partition model (Hartigan, 1990) makes use of a manually specified partitioning of the $D$ variables into subsets

$$\{1, \ldots, D\} = \bigcup_{\ell=1}^{L} S_\ell.$$

For each subset of variables $\boldsymbol{x}(S_\ell) = (\boldsymbol{x}(d))_{d \in S_\ell}$ there exists a separate model $\mathbb{P}_\ell$. It is then typically assumed that variables in different subsets are conditionally independent, i.e.,

$$\mathbb{P}(\boldsymbol{x} \mid \boldsymbol{h}) = \prod_{\ell=1}^{L} \mathbb{P}_\ell(\boldsymbol{x}(S_\ell) \mid \boldsymbol{h}(\ell)). \tag{1}$$

The model is completed by specifying a prior distribution $\mathbb{P}(\boldsymbol{h})$ for the latent state $\boldsymbol{h}$. One advantage of partition models is that estimating $\mathbb{P}_\ell$ from observations is straightforward, while learning expert models in general requires computationally involved procedures (Bengio et al., 2013). However, in order to be able to define a satisfactory partitioning of the variables some prior knowledge about the dependence structure is needed. For image data a common choice is to use a regular grid that divides the image into patches (e.g. Pal et al., 2002). In general, a good partitioning is characterized by providing weakly dependent subsets of variables so that the conditional independence assumption (1) is reasonable and the distribution of the latent variables is easy to model. Unfortunately, often there simply is no single fixed partitioning that works well for the whole dataset because the set

of variables, which are affected by different factors of variation, might overlap. This restricts the scenarios in which partition models are useful.

In this paper we extend partition models to allow for dynamically adapting partitionings. In Section 2 we introduce the model and present an appropriate learning procedure. Related work is discussed in Section 3. Special emphasis is given to the comparison with products of experts (Hinton, 2002). Experiments on binary and real-valued data are performed in Section 4. While it is important to explain high-dimensional data points through multiple experts, our work shows that it is possible to assign the responsibility for individual variables to a single expert (rather than having all active experts speak for every variable).

## 2 DYNAMIC PARTITION MODELS

Our main proposal is to define for each expert $\mathbb{P}_k$ its level of expertise $\boldsymbol{e_k} \in \mathbb{R}_+^D$ for all variables. We can then dynamically partition the variables based on the active experts. Specifically, for each variable we employ the most reliable (active) expert

$$\mathbb{P}(\boldsymbol{x} \,|\, \boldsymbol{h}) = \prod_{d=1}^{D} \mathbb{P}_{k^\star(d)}(\boldsymbol{x}(d)), \qquad k^\star(d) = \operatorname*{argmax}_{k:\boldsymbol{h}(k)=1} \boldsymbol{e_k}(d). \tag{2}$$

That means, each variable $\boldsymbol{x}(d)$ is explained by only a single expert $k^\star(d)$. The partitioning into expert supports $S_k(\boldsymbol{h}) = \{d \in \{1, \ldots, D\} : k^\star(d) = k\}$ is determined dynamically based on the latent configuration $\boldsymbol{h}$. We hence call our model a dynamic partition model.

### 2.1 INFERENCE

In the inference step we try to find for each data point $\boldsymbol{x_n}$ the subset of experts $\{k : \boldsymbol{h_n}(k) = 1\}$ that maximizes $P(\boldsymbol{x_n} \,|\, \boldsymbol{h_n})$. To do this, we suggest to sequentially activate the expert that most improves the likelihood, until the likelihood cannot be improved anymore. This approach is called likelihood matching pursuit (Goessling & Amit, 2015). The greedy search works well for our model because we are working with a small set of experts and each expert focuses on a rather different structure in the data. Consequently, the posterior distribution on the latent variables given $\boldsymbol{x_n}$ is often highly peaked at a state $\boldsymbol{h_n}$ (note that for high-dimensional data the effect of the prior $\mathbb{P}(\boldsymbol{h})$ is typically negligible).

### 2.2 LEARNING

In contrast to traditional approaches, which combine multiple experts for individual variables, training the experts in a dynamic partition model is trivial. Indeed, the maximum-likelihood estimates are simply the empirical averages over all observations for which the expert was responsible. For example, the expert means can be estimated from training data $\boldsymbol{x_n}, n = 1, \ldots, N$, as

$$\overset{\circ}{\boldsymbol{\mu_k}}(d) = \frac{\sum_{n=1}^{N} \mathbb{1}\{k_n^\star(d)=k\}\boldsymbol{x_n}(d)}{\sum_{n=1}^{N} \mathbb{1}\{k_n^\star(d)=k\}}. \tag{3}$$

Here, $k_n^\star(d)$ denotes the expert with the highest level of expertise $\boldsymbol{e_k}(d)$ among all experts $k$ with $\boldsymbol{h_n}(k) = 1$.

#### 2.2.1 EXPERTISE-WEIGHTED COMPOSITION

In order to compute the estimator in (3) the levels of expertise $\boldsymbol{e_k}$ have to be known. Since in this paper we are trying to train the experts as well as the associated levels of expertise we consider a smoothing of the maximum-expertise composition (2) to motivate our learning procedure. Rather than using the expert with the highest level of expertise, we form a mixture of the active experts, where the mixture weight is proportional to the level of expertise. Thus, the smoothed composition

rule is

$$\widetilde{\mathbb{P}}(\boldsymbol{x} \mid \boldsymbol{h}) = \prod_{d=1}^{D} \sum_{k=1}^{K} \boldsymbol{r_k}(d) \mathbb{P}_k(\boldsymbol{x}(d)), \qquad \boldsymbol{r_k}(d) = \begin{cases} \frac{\boldsymbol{e_k}(d)}{\sum_{k':\boldsymbol{h}(k')=1} \boldsymbol{e_{k'}}(d)} & \text{if } \boldsymbol{h}(k) = 1 \\ 0 & \text{if } \boldsymbol{h}(k) = 0 \end{cases}. \tag{4}$$

In contrast to classical mixture models (e.g. McLachlan & Peel, 2004) we use different mixture weights for each dimension $d \in \{1, \ldots, D\}$. The mixture weight $\boldsymbol{r_k}(d)$ is the degree of responsibility of $k$-th expert for the $d$-th dimension and depends on the latent state $\boldsymbol{h}$. An expert with a medium level of expertise assumes full responsibility if no other reliable expert is present and takes on a low degree of responsibility if experts with a higher level of expertise are present.

According to the total variance formula

$$\mathbb{V}[\widetilde{\mathbb{P}}] = \mathbb{E}_{\boldsymbol{r_k}}[\mathbb{V}[\mathbb{P}_k]] + \mathbb{V}_{\boldsymbol{r_k}}[\mathbb{E}[\mathbb{P}_k]]$$

the variance of a mixture is always larger than the smallest variance of its components. In other words, the precision of the smoothed model is maximized when all the mixture weight (individually for each dimension) is concentrated on the most precise expert. We can thus learn a dynamic partition model in an EM manner (Dempster et al., 1977) by interleaving inference steps with updates of the experts and levels of expertise in the smoothed model.

### 2.2.2 EXPERT UPDATE

The sequential inference procedure (from Section 2.1) provides for each data point $\boldsymbol{x_n}$ the latent representation $\boldsymbol{h_n}$. We denote the corresponding expert responsibilities (using the current estimates for the level of expertise) by $\boldsymbol{r_{nk}}$. The smooth analog to the hard update equation (3) is a responsibility-weighted average of the training samples

$$\boldsymbol{\mu_k}(d) = \frac{\sum_{n=1}^{N} \boldsymbol{r_{nk}}(d) \boldsymbol{x_n}(d) + \epsilon \boldsymbol{\mu_0}}{\sum_{n=1}^{N} \boldsymbol{r_{nk}}(d) + \epsilon}. \tag{5}$$

For stability we added a term that shrinks the updated templates towards some target $\boldsymbol{\mu_0}$ if the total responsibility of the expert is small. In our experiments we set $\boldsymbol{\mu_0}$ to the average of all training examples. The update rule implies that the experts have local supports, in the sense that they are uninformative about variables for which they are not responsible.

For binary data the mean templates $\boldsymbol{\mu_k}$ are all we need. Continuous data $\boldsymbol{x} \in \mathbb{R}^D$ is modeled through Gaussians and hence we also have to specify the variance $\boldsymbol{v_k}$ of the experts. We again use a responsibility-weighted average

$$\boldsymbol{v_k}(d) = \frac{\sum_{n=1}^{N} \boldsymbol{r_{nk}}(d)(\boldsymbol{x_n}(d) - \boldsymbol{\mu_k}(d))^2 + \epsilon \boldsymbol{v_0}}{\sum_{n=1}^{N} \boldsymbol{r_{nk}}(d) + \epsilon}, \tag{6}$$

where $\boldsymbol{v_0}$ is the empirical variance of all training samples.

### 2.2.3 EXPERTISE UPDATE

We now turn to the updates of the levels of expertise. The log-likelihood of the smoothed model (4) as a function of $\boldsymbol{e}_k$ is rather complex. Using gradient descent is thus problematic because the derivatives with respect to $\boldsymbol{e_k}$ can have very different scales, which makes it difficult to choose an appropriate learning rate and hence the convergence could be slow. However, exact optimization is not necessary because in the end only the order of the levels of expertise matters. Consequently, we propose to adjust $\boldsymbol{e_k}(d)$ only based on the sign of the gradient. We simply multiply or divide the current value by a constant $C$. If the gradient is very close to $0$ we leave $\boldsymbol{e_k}(d)$ unchanged. For all our experiments we used $C = 2$. Larger values can speed up the convergence but sometimes lead to a worse solution. Using an exponential decay is common practice when learning levels of expertise (e.g. Herbster & Warmuth, 1998).

In the learning procedure we perform the expertise update first. We then recompute the responsibilities using these new levels of expertise and update the experts. Our algorithm typically converges after about 10 iterations.

## 3 RELATED WORK

Herbster & Warmuth (1998) proposed an algorithm for tracking the best expert in a sequential prediction task. In their work it is assumed that a linear ordering of the variables is known such that the expert with the highest level of expertise is constant on certain segments. In contrast to that, our approach can be applied to an arbitrary permutation of the variables. Moreover, they consider a single sequence of variables with a fixed partitioning into experts supports. In our setup the partitioning changes dynamically depending on the observed sample. However, the greatest difference to our work is that Herbster & Warmuth (1998) do not learn the individual experts but only focus on training the levels of expertise.

Lücke & Sahani (2008) studied a composition rule that also partitions the variables into expert supports. In their model the composed template is simply the maximum of the experts templates $\boldsymbol{\mu_k}$. This rule is only useful in special cases. A generalization, in which the composition depends on the maximum and the minimum of the expert templates $\boldsymbol{\mu_k}(d)$, was considered by Goessling & Amit (2015). While the motivation for that rule was similar, the maximum-expertise rule in this paper is more principled and can be applied to continuous data.

In the work by Amit & Trouvé (2007) a simple average (i.e., an equal mixture) of the individual templates was used. With such a composition rule, all experts are equally responsible for each of the variables and hence specialization on local structures is not possible. To circumvent this problem, in their work $\boldsymbol{e_k}(d)$ was manually set to 1 for some subset of the dimensions (depending on a latent shift variable) and to 0 elsewhere.

A popular model family with latent binary representation are products of experts (Hinton, 2002). In such a model the individual distributions $\mathbb{P}_k$ are multiplied together and renormalized. Computation of the normalizing constant is in general intractable though. A special case, in which an explicit normalization is possible, are restricted Boltzmann machines (Hinton, 2002). In these models the experts are product Bernoulli distributions with templates $\boldsymbol{\mu_k} \in [0,1]^D$. The composed distribution is then also a product Bernoulli distribution with composed template

$$\boldsymbol{\mu}_{\text{RBM}}(d) = \sigma\left(\sum\nolimits_{k:\boldsymbol{h}(k)=1} \boldsymbol{w_k}(d)\right),$$

where the weights $\boldsymbol{w_k}(d) = \log(\boldsymbol{\mu_k}(d)/(1 - \boldsymbol{\mu_k}(d)) \in \mathbb{R}$ are the log-odds of the experts and $\sigma(t) = (1 + \exp(-t))^{-1}$ is the logistic function. This sum-of-log-odds composition rule arises naturally from generalized linear models for binary data because the log-odds are the canonical parameter of the Bernoulli family. In a product of experts, the variance of the composition is usually smaller than the smallest variance of the experts. As a consequence, products of experts tend to employ many experts for each dimension (for more details on this issue see Goessling & Amit (2015)). Even with an L1-penalty on the votes $\boldsymbol{w_k}(d)$ the responsibility for individual variables $\boldsymbol{x}(d)$ is typically still shared among many experts. The reason for this is that under the constraint $\sum_k \boldsymbol{w_k}(d) = \boldsymbol{w}(d)$ the quantity $\sum_k |\boldsymbol{w_k}(d)|$ is minimized whenever $\boldsymbol{w_k}(d)$ has the same sign for all $k$. The usual inference procedure for products of experts independently activates experts based on their inner product with the data point. In particular, not just the most probable expert configuration is determined but the whole posterior distribution on latent states given the data is explored through Monte Carlo methods. For learning in products of experts, simple update rules like (5) and (6) cannot be used because for each expert the effects of all other experts have to be factored out. Dynamic partition models essentially decompose the expert votes $\boldsymbol{w_k}$ into expert opinions $\boldsymbol{\mu_k}$ and levels of expertise $\boldsymbol{e_k}$. Apart from the computational advantages for learning, this introduces an additional degree of flexibility because the expert supports are adjusted depending on which other experts are present (cf. Figure 5). Moreover, the decomposition into opinions and levels of expertise avoids ambiguities. For example, a vote $\boldsymbol{w_k}(d) \approx 0$ could mean that $\boldsymbol{\mu_k}(d) \approx 1/2$ or that $\boldsymbol{e_k}(d) \approx 0$.

Another common model for representation learning are autoencoders (Vincent et al., 2008), which can be considered as mean-field approximations of restricted Boltzmann machines that use latent variables $\boldsymbol{h}(k)$ with values in $[0,1]$. To obtain a sparse representation a penalty on the number of active experts can be added (Ng, 2011). Such approaches are also known as sparse dictionaries (e.g., Elad, 2010) and are based on opinion pools of the form $\sum_k \boldsymbol{h}(k)\boldsymbol{w_k}(d)$. The strength of the sparsity penalty is an additional tuning parameter which has to be tuned. In dynamic partition models sparse activations are inherent. In the next section, we experimentally compare products of experts, autoencoders and sparse dictionaries to our proposed model.

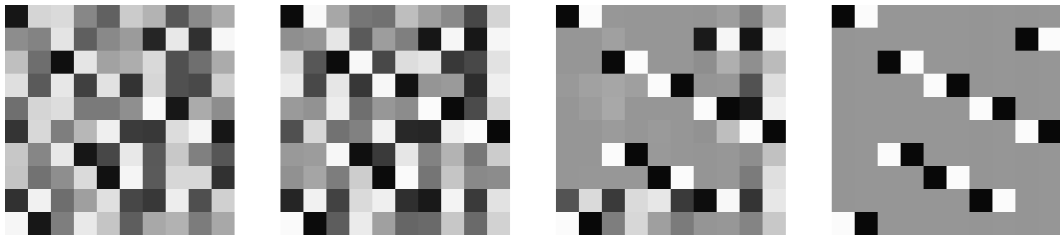

Figure 1: Expert training for the synthetic dataset. Each panel shows the probabilities (white/black corresponds to $\boldsymbol{\mu_k}(d) = 0/1$) of the 10 experts (rows) for the 10 dimensions (columns). **1st panel:** Random initialization. **2nd-4th panel:** Our learning procedure after 3/5/15 iterations.

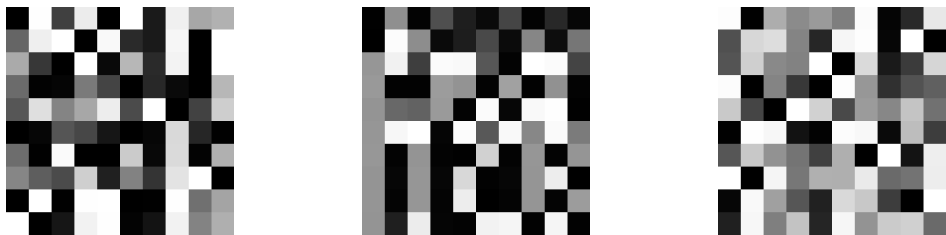

Figure 2: Trained experts for the synthetic data after 1,000 iterations using an autoencoder (**1st panel**), a sparse dictionary (**2nd panel**) and a restricted Boltzmann machine (**3rd panel**).

## 4 EXPERIMENTS

### 4.1 SYNTHETIC DATA

We consider a synthetic example and try to learn the underlying factors of variation. The dataset consists of the 32-element subset $\{(0, 1), (1, 0)\}^5 \subset \{0, 1\}^{10}$. Note that there are 5 factors of variation corresponding to the state of the pairs $(\boldsymbol{x}(2\ell-1), \boldsymbol{x}(2\ell))$ for $\ell = 1, \dots, 5$ with the two factor levels $(0, 1)$ and $(1, 0)$. Indeed, the distribution can be easily expressed through a partition model with partitioning

$$\{1, 2\} \cup \{3, 4\} \cup \{5, 6\} \cup \{7, 8\} \cup \{9, 10\}$$

and corresponding models

$$\mathbb{P}_\ell(\boldsymbol{x}(2\ell-1), \boldsymbol{x}(2\ell)) = \tfrac{1}{2} \cdot \mathbb{1}\{\boldsymbol{x}(2\ell-1)=0,\ \boldsymbol{x}(2\ell)=1\} + \tfrac{1}{2} \cdot \mathbb{1}\{\boldsymbol{x}(2\ell-1)=1,\ \boldsymbol{x}(2\ell)=0\}.$$

We show that our dynamic partition model is able to learn these factors of variation without requiring a manual specification of the partitioning. Here, the total number of experts we need to accurately reconstruct all data points happens to be equal to the number of dimensions. However, in other cases the number of required experts could be smaller or larger than $D$. We ran our learning algorithm for 15 iterations starting from a random initialization of the experts. The resulting templates after 3, 5 and 15 iterations are shown in Figure 1. We see that each of the final experts specializes in exactly two dimensions $d$ and $d+1$. Its opinion for these variables are close to 0 and 1, respectively, while the opinions for the remaining variables are about 1/2. Every data point can now be (almost) perfectly reconstructed by using exactly 5 of these experts.

For comparison we trained various other models with 10 experts, which use a sum-of-log-odds composition. We first tried an autoencoder (Vincent et al., 2008), which in principle could adopt the identity map because it uses (in contrast to our model) a bias term for the observable and latent variables. However, the gradient descent learning algorithm with tuned step size yielded a different representation (Figure 2, 1st panel). While the reconstruction errors are rather low, they are clearly nonzero and the factors of variations have not been disentangled. Next, we considered a dictionary with a sparse representation (e.g., Elad, 2010). The sparsity penalty was adjusted so that the average number of active dictionary elements was around 5. The learning algorithm again yielded highly dependent experts (Figure 2, 2nd panel). Finally, we trained a restricted Boltzmann machine through batch persistent contrastive divergence (Tieleman, 2008) using a tuned learning rate. Note that a

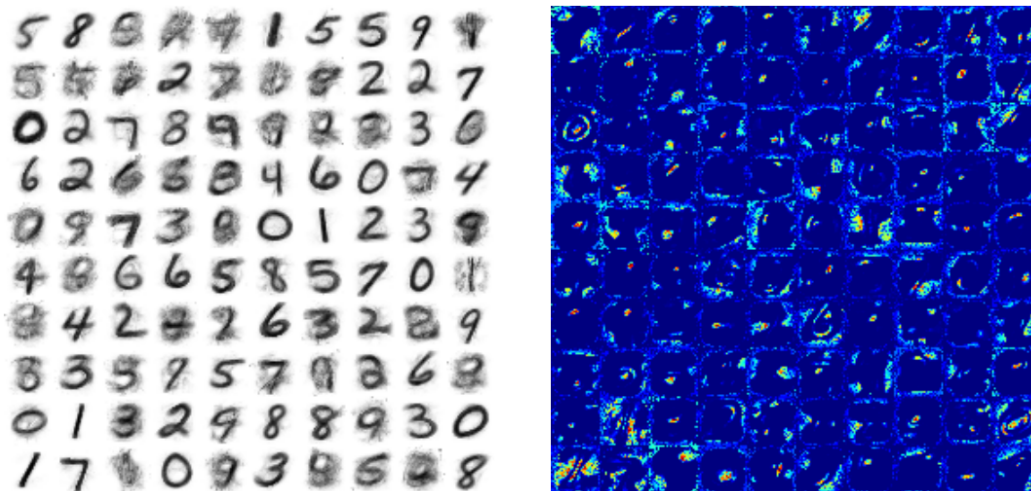

Figure 3: Trained experts for MNIST digits. **Left:** Expert probabilities (white/black corresponds to $\boldsymbol{\mu_k}(d) = 0/1$). **Right:** Levels of expertise (blue/red corresponds to small/large values).

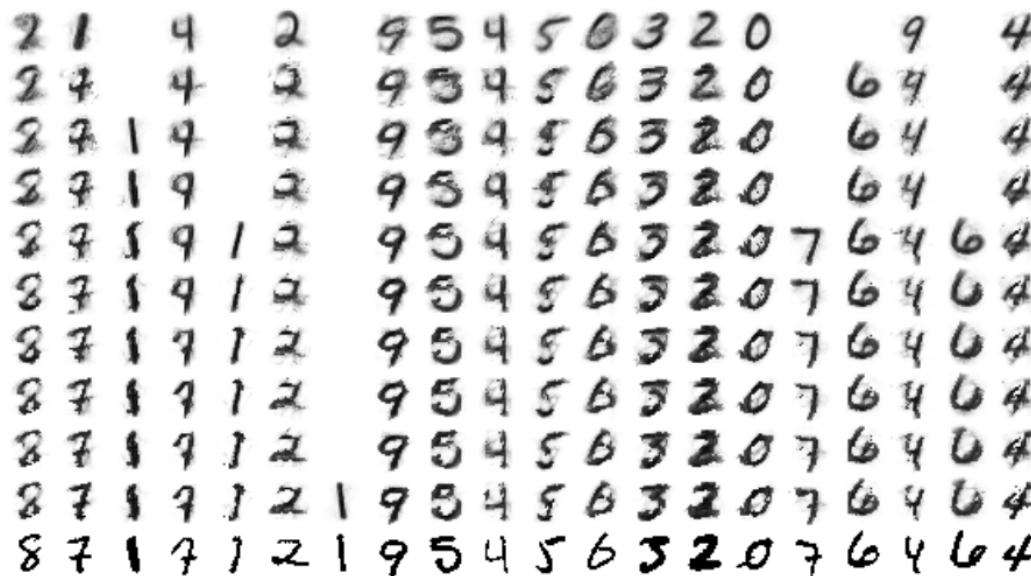

Figure 4: Reconstruction of MNIST test examples using likelihood matching pursuit. Each column visualizes the composed Bernoulli templates during the sequential inference procedure (top down) for one sample. The bottom row are the original data points.

restricted Boltzmann machine in principle only requires 5 experts to model the data appropriately because it uses bias terms. However, we again learned 10 experts (Figure 2, 3rd panel). While the results look better than for the previous two models they are still far from optimal. In earlier work Goessling & Amit (2015) we performed a quantitative comparison for a similar dataset, which showed that the reconstruction performance of models with sum-of-log-odds composition is indeed suboptimal.

## 4.2 MNIST DIGITS

We now consider the MNIST digits dataset (LeCun et al., 1998), which consists of 60,000 training samples and 10,000 test samples of dimension $28 \times 28 = 784$. We ran our learning algorithm for 10

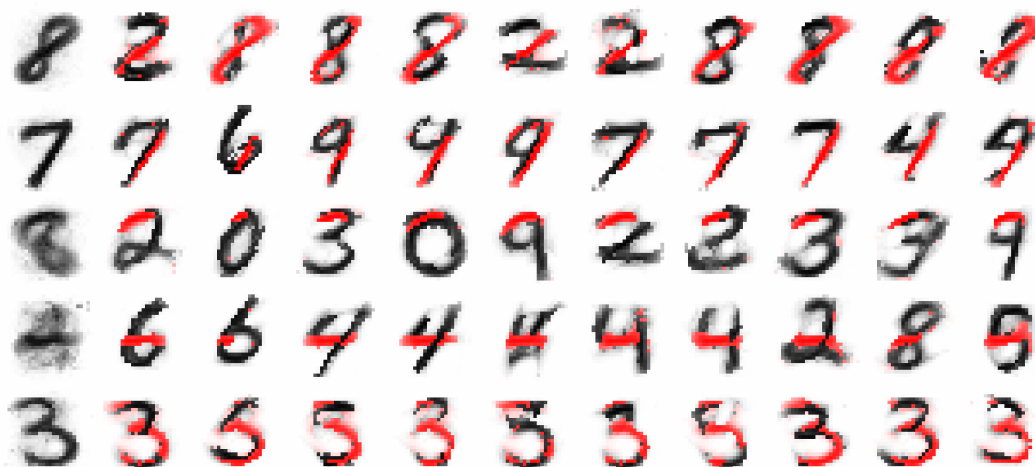

Figure 5: Dynamic supports for 5 MNIST experts. **Left column:** Expert probabilities. **Remaining columns:** Composed Bernoulli templates for 10 latent configurations. The cast opinion of the expert is shown in shades of red (white/red corresponds to $\boldsymbol{\mu_k}(d) = 0/1$).

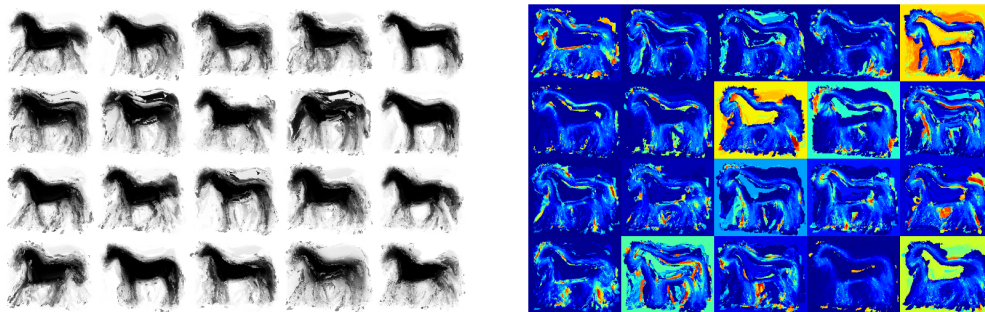

Figure 6: Trained experts for Weizmann horses. **Left:** Expert probabilities (white/black corresponds to $\boldsymbol{\mu_k}(d) = 0/1$). **Right:** Levels of expertise (blue/red corresponds to small/large values).

iterations and trained 100 experts (Figure 3). We see that some experts specialize on local structures while others focus on more global ones. In Figure 4 we visualize the inference procedure for some test samples using these 100 learned experts. On average 12 experts were activated for each data point. For easier visualization we show at most 10 iterations of the likelihood matching pursuit algorithm. The reconstructions are overall accurate and peculiarities of the samples are smoothed out. In Figure 5 we illustrate how the expert supports change based on the latent representation. Depending on which other experts are present the supports can vary quite a bit.

### 4.3 WEIZMANN HORSES

The following experiment shows that our model is able to cope with very high-dimensional data. The Weizmann horse dataset (Borenstein & Ullman, 2008) consists of 328 binary images of size $200 \times 240$. We used the first 300 images to train 20 experts (Figure 6) and used the remaining 28 images for testing. Some of the experts are responsible for the background and the central region of the horse while other experts focus on local structures like head posture, legs and tail. In Figure 7 we illustrate the partitioning of the test examples into expert opinions. For simplicity we used exactly 4 experts to reconstruct each sample. Not all characteristics of the samples are perfectly reconstructed but the general pose is correctly recovered. The same dataset was used to evaluate the shape Boltzmann machine (Eslami et al., 2014), where 2,000 experts were learned. For those experiments the images were downsampled to $32 \times 32$ pixels. This is a factor 50 smaller than the full resolution of 48,000 dimensions that we use.

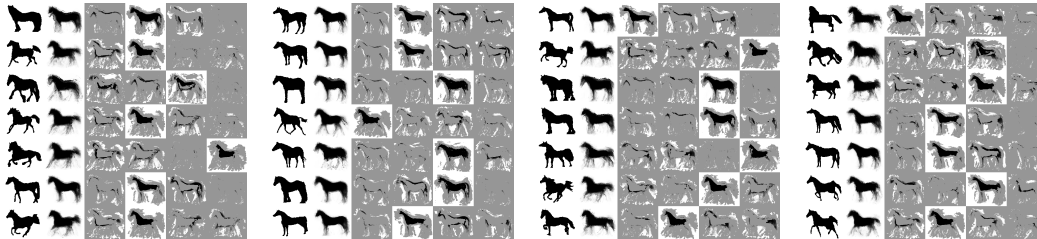

Figure 7: Decomposition of the test examples from the Weizmann horse dataset. **1st column:** Original data points. **2nd column:** Reconstructions (shown are the composed Bernoulli templates). **3rd-6th column:** Partitioning into experts opinions (white/black corresponds to $\boldsymbol{\mu_k}(d) = 0/1$, gray indicates regions for which the expert is not responsible).

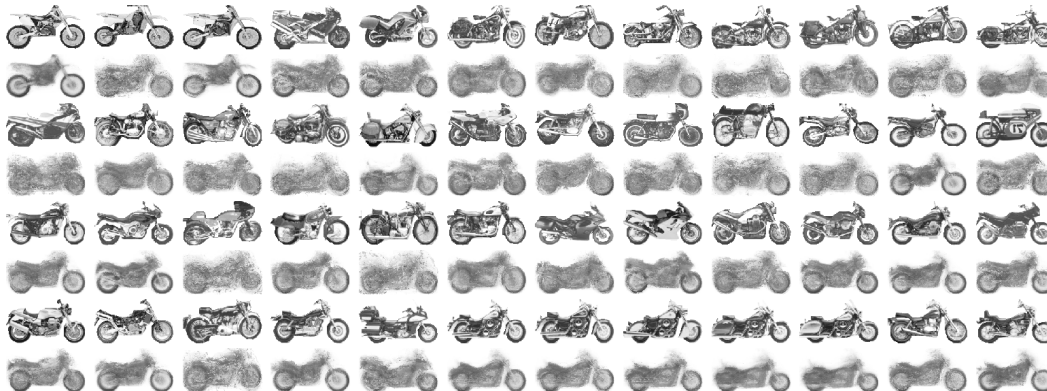

Figure 8: Reconstructions of the test examples from the Caltech motorcycle dataset. **Odd rows:** Original data. **Even rows:** Reconstructions (shown are the composed Gaussian means).

## 4.4 CALTECH MOTORCYCLES

We also experimented with real-valued data using the Caltech-101 motorcycle dataset (Fei-Fei et al., 2007), which consists of 798 images of size $100 \times 180$. The first 750 images were used for training and the remaining 48 images for testing. We trained 50 experts by running our learning procedure for 10 iterations. In Figure 8 we visualize the reconstructed test examples. The reconstructions are a bit blurry since we use a fairly sparse binary representation. Indeed, for each data point on average only 7 experts were employed. Note that the shapes of the motorcycles are reconstructed quite accurately.

## 5 DISCUSSION

In order to improve the reconstructions for continuous image data we could use real-valued latent variables in addition to binary ones (as in Hinton et al. (1998)). This would allow us to model intensities and contrasts more accurately. The inference procedure would have to be adapted accordingly such that continuous activations can be returned.

Our work focused on product distributions. In order to apply the proposed approach to models with dependence structure one can make use of an autoregressive decomposition (e.g., Goessling & Amit, 2016). If the joint distribution is written as a product of conditional distributions then we can employ the same composition rule as before. Indeed, we can model composed the conditionals as

$$\mathbb{P}(\boldsymbol{x}(d) \,|\, \boldsymbol{x}(1{:}d{-}1), \boldsymbol{h}) = \mathbb{P}_{k^{\star}(d)}(\boldsymbol{x}(d) \,|\, \boldsymbol{x}(1{:}d{-}1)),$$

where $\mathbb{P}_k$ are autoregressive expert models and $k^{\star}(d)$ is the active expert with the highest level of expertise for dimension $d$.

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

## 6 DERIVATIVES

We provide here the derivatives of the log-likelihood in the expertise-weighted compositional model (4) with respect to the expert parameters.

### 6.1 BERNOULLI MODEL

The Bernoulli log-likelihood is

$$f(\mu) = x \log \mu + (1 - x) \log(1 - \mu),$$

where the composition rule for the probability is

$$\mu = \sum_k r_k \mu_k, \quad r_k = \frac{e_k}{\sum_{k'} e_{k'}}.$$

#### 6.1.1 DERIVATIVES WITH RESPECT TO THE COMPOSED PROBABILITY

The first and second derivative of the log-likelihood with respect to the composed probability are

$$\frac{df}{d\mu} = \frac{x}{\mu} - \frac{1 - x}{1 - \mu} = \frac{x - \mu}{\mu(1 - \mu)},$$

$$\frac{d^2 f}{d\mu^2} = -\frac{x}{\mu^2} - \frac{1 - x}{(1 - \mu)^2} = -\frac{(x - \mu)^2}{\mu^2(1 - \mu)^2}.$$

#### 6.1.2 DERIVATIVES WITH RESPECT TO THE EXPERT PROBABILITIES

The first and second derivative of the composed probability with respect to the expert probabilities are

$$\frac{d\mu}{d\mu_k} = r_k, \quad \frac{d^2\mu}{d\mu_k^2} = 0.$$

Consequently, the derivatives of the log-likelihood with respect to the expert probabilities are

$$\frac{df}{d\mu_k} = \frac{df}{d\mu} \cdot \frac{d\mu}{d\mu_k} = r_k \frac{x - \mu}{\mu(1 - \mu)},$$

$$\frac{d^2 f}{d\mu_k^2} = \frac{d^2 f}{d\mu^2} \cdot \left(\frac{d\mu}{d\mu_k}\right)^2 + \frac{df}{d\mu} \cdot \frac{d^2\mu}{d\mu_k^2} = -r_k^2 \frac{(x - \mu)^2}{\mu^2(1 - \mu)^2}.$$

We see that $d^2 f/d\mu_k^2 < 0$ for $\mu \in (0, 1)$, i.e., the log-likelihood is a strictly concave function of $\mu_k$.

#### 6.1.3 DERIVATIVE WITH RESPECT TO THE LEVELS OF EXPERTISE

The derivative of the composed probability with respect to the levels of expertise is

$$\frac{d\mu}{de_k} = \frac{\mu_k E - \sum e_{k'} \mu_{k'}}{E^2} = \frac{\mu_k - \mu}{E},$$

where $E = \sum_{k'} e_{k'}$. The derivative of the log-likelihood with respect to the levels of expertise can be computed as

$$\frac{df}{de_k} = \frac{df}{d\mu} \cdot \frac{d\mu}{de_k}.$$

### 6.2 GAUSSIAN MODEL

The Gaussian log-likelihood is

$$f(\mu, v) = -\frac{(x - \mu)^2}{2v} - \frac{1}{2} \log(v) - \frac{1}{2} \log(2\pi),$$

where the composition rules for the mean and variance are

$$\mu = \sum_k r_k \mu_k, \quad v = \sum_k r_k (v_k + \mu_k^2) - \mu^2, \quad r_k = \frac{e_k}{\sum_{k'} e_{k'}}.$$

### 6.2.1 DERIVATIVE WITH RESPECT TO THE COMPOSED MEAN AND VARIANCE

The derivative of the log-likelihood with respect to the composed mean and variance are

$$\frac{df}{d\mu} = \frac{x - \mu}{v}, \quad \frac{df}{dv} = \frac{(x - \mu)^2}{2v^2} - \frac{1}{2v} = \frac{(x - \mu)^2 - v}{2v^2}.$$

### 6.2.2 DERIVATIVE WITH RESPECT TO THE LEVELS OF EXPERTISE

The derivative of the composed mean and variance with respect to the levels of expertise are

$$\frac{d\mu}{de_k} = \frac{\mu_k E - \sum e_{k'} \mu_{k'}}{E^2} = \frac{\mu_k - \mu}{E},$$

$$\frac{dv}{de_k} = \frac{q_k E - \sum e_{k'} q_{k'}}{E^2} - 2\mu \frac{d\mu}{de_k} = \frac{q_k - q}{E} - 2\mu \frac{\mu_k - \mu}{E} = \frac{v_k - v + (\mu_k - \mu)^2}{E},$$

where $E = \sum_{k'} e_{k'}$ and $q_k = v_k + \mu_k^2, q = v + \mu^2$. The derivative of the log-likelihood with respect to the levels of expertise can be computed as

$$\frac{df}{de_k} = \frac{df}{d\mu} \cdot \frac{d\mu}{de_k} + \frac{df}{dv} \cdot \frac{dv}{de_k}.$$

## 7 NUMERICAL OPTIMIZATION

For binary data, the log-likelihood of the smoothed model is a concave function of $\boldsymbol{\mu_k}(d)$, see Section 6.1.2. We could therefore in principal perform an optimization for the experts opinions using Newton's method. There are a few complications though. One problem is that the second derivative is proportional to the squared responsibility and hence close to 0 if the level of expertise is small. Consequently, template updates in regions with low expertise would be unstable. To deal with that we could add a penalty on the squared log-odds for example. Another problem is that the Newton steps may lead to probability estimates outside of $[0, 1]$. This can be dealt with by pulling the estimates back into the unit interval. Note that working on the log-odds scale is not possible because the log-likelihood of our model is not concave in the expert log-odds. Because of these complications we use the simple, fast and robust heuristic (5) instead of Netwon's method.

