# Peer review of "Dynamic Partition Models"

_ICLR 2017 — rejected_

[Official Review · AnonReviewer3 · rating 3 · confidence 4 · 14 Dec 2016]
**A type of PoE but the probability seems undefined and the EM algorithms remains obscure. Experiments are illustrative only.**

This paper proposes a new kind of expert model where a sparse subset of most reliable experts is chosen instead of the usual logarithmic opinion pool of a PoE.
I find the paper very unclear. I tried to find a proper definition of the joint model p(x,z) but could not extract this from the text. The proposed “EM-like” algorithm should then also follow directly from this definition. At this point I do not see if such as definition even exists. In other words, is there is an objective function on which the iterates of the proposed algorithm are guaranteed to improve on the train data?
We also note that the “product of unifac models” from Hinton tries to do something very similar where only a subset of the experts will get activated to generate the input:

[Official Review · AnonReviewer2 · rating 6 · confidence 3 · 16 Dec 2016]
**Improve the exposition**

The goal of this paper is to learn “ a collection of experts that are individually
meaningful and that have disjoint responsibilities.” Unlike a standard mixture model, they “use a different mixture for each dimension d.” While the results seem promising, the paper exposition needs significant improvement.

Comments:

The paper jumps in with no motivation at all. What is the application, or even the algorithm, or architecture that this is used for? This should be addressed at the beginning.

The subsequent exposition is not very clear. There are assertions made with no justification, e.g. “the experts only have a small variance for some subset of the variables while the variance of the other variables is large.” 

Since you’re learning both the experts and the weights, can this be rephrased in terms of dictionary learning? Please discuss the relevant related literature.

The horse data set is quite small with respect to the feature dimension, and so the conclusions may not necessarily generalize.

[Official Review · AnonReviewer1 · rating 3 · confidence 4 · 17 Dec 2016]
**Potentially interesting paper, but not clear enough**

The paper addresses the problem of learning compact binary data representations. I have a hard time understanding the setting and the writing of the paper is not making it any easier. For example I can't find a simple explanation of the problem and I am not familiar with these line of research. I read all the responses provided by authors to reviewer's questions and re-read the paper again and I still do not fully understand the setting and thus can't really evaluate the contributions of these work. The related work section does not exist and instead the analysis of the literature is somehow scattered across the paper. There are no derivations provided. Statements often miss references, e.g. the ones in the fourth paragraph of Section 3. This makes me conclude that the paper still requires significant work before it can be published.

[Final Decision · Program Chairs · 06 Feb 2017]
**ICLR committee final decision**

This paper is about learning distributed representations. All reviewers agreed that the first draft was not clear enough for acceptance.
 
 Reviewer time is limited and a paper that needed a complete overhaul after the reviews were written is not going to get the same consideration as a paper that was well-drafted from the beginning.
 
 It's still the case that it's unclear from the paper how the learning updates or derived. The results are not visually impressive in themselves. It's also still the case that more is needed to demonstrate that this direction is promising compared to other approaches to representation learning.